# Calcareous Materials Effectively Reduce the Accumulation of Cd in Potatoes in Acidic Cadmium-Contaminated Farmland Soils in Mining Areas

**DOI:** 10.3390/ijerph191811736

**Published:** 2022-09-17

**Authors:** Sitong Gong, Hu Wang, Fei Lou, Ran Qin, Tianling Fu

**Affiliations:** 1Guizhou University, Guiyang 550025, China; 2Guizhou Chuyang Ecological Environmental Protection Technology Co., Ltd., Guiyang 550025, China

**Keywords:** Cadmium (Cd), potato, passivation, bioavailability, morphology

## Abstract

The in situ chemical immobilization method reduces the activity of heavy metals in soil by adding chemical amendments. It is widely used in farmland soil with moderate and mild heavy metal pollution due to its high efficiency and economy. However, the effects of different materials depend heavily on environmental factors such as soil texture, properties, and pollution levels. Under the influence of lead–zinc ore smelting and soil acidification, Cd is enriched and highly activated in the soils of northwestern Guizhou, China. Potato is an important economic crop in this region, and its absorption of Cd depends on the availability of Cd in the soil and the distribution of Cd within the plant. In this study, pot experiments were used to compare the effects of lime (LM), apatite (AP), calcite (CA), sepiolite (SP), bentonite (BN), and biochar (BC) on Cd accumulation in potatoes. The results showed that the application of LM (0.4%), AP (1.4%), and CA (0.4%) had a positive effect on soil pH and cations, and that they effectively reduced the availability of Cd in the soil. In contrast, the application of SP, BN, and BC had no significant effect on the soil properties and Cd availability. LM, AP, and CA treatment strongly reduced Cd accumulation in the potato tubers by controlling the total ‘flux’ of Cd into the potato plants. In contrast, the application of SP and BN promoted the migration of Cd from the root to the shoot, while the effect of BC varied by potato genotype. Overall, calcareous materials (LM, CA, and AP) were more applicable in the remediation of Cd-contaminated soils in the study area.

## 1. Introduction

Soil is one of the natural resources essential for human survival, the medium for plant growth, and the buffer zone and filter for environmental pollutants. However, when environmental pollutants entering the soil exceed the environmental capacity, it may cause soil pollution and endanger human health. Once soil is contaminated with cadmium (Cd), Cd can accumulate in crops and enter the human body through the food chain, which in turn causes kidney disease, bone damage, and even cancer [1,2]. The concentrations of relevant heavy metals (i.e., arsenic, cadmium, chromium, copper, lead, mercury, and zinc) in Chinese soils have gradually increased over the past 20 years, while Cd and Hg are the most dominating polluting elements [3]. Cd is a high-risk element that needs to be controlled as a priority, especially in agricultural areas [4].

Currently, in situ- and ex situ-remediation techniques such as soil drenching, electro-extraction, immobilization, vitrification, phyto- and microbial-remediation have been widely used in the remediation of heavy-metal-contaminated soil [5]. The in situ chemical immobilization method reduces the activity of heavy metals in soil by adding chemical amendments, which in turn reduces the migration of heavy metals to plants. Due to its high efficiency and economy, the in situ chemical immobilization method can inhibit the biotoxicity of heavy metals without affecting normal agricultural production, and has the potential for a wide range of applications in farmland soils polluted with heavy metals [5]. The application of chemical amendments can reduce the availability of Cd in soil by the adsorption mechanism, precipitation mechanism, redox mechanism, and redox mechanism, which in turn inhibits the absorption of Cd by plants [6,7,8,9].

However, even if in situ chemical immobilization methods have been widely used in soil improvement, their Cd-reducing ability in crops is often inconsistent. The remediation effect of different materials is largely dependent on environmental factors such as soil texture, properties, and pollution levels. For example, a meta-analysis of the effects of lime application showed that the type of experiment (pot or field), dosage, duration, soil properties, and crop cultivar all affected the effectiveness of lime to remediate soil. Among them, the effects of soil pH, total Cd concentration, and crop varieties cannot be ignored [10]. Another study showed that soil pH, organic matter (SOM), cation exchange capacity (CEX), and soil texture perhaps determines the remediation effect of lime, and that lower soil pH, SOM, CEC, and clay may be beneficial in improving the remediation effect [11]. Similarly, a meta-analysis of the effects of biochar application showed that soil properties (such as pH, soil organic carbon, and texture) largely affected the effect of biochar [12]. The results from the simulation experiments also showed that NaOH-modified biochar (CSB-NaOH) was more effective than CSB in regulating Cd bioavailability in acidic soils, while the opposite trend was observed in alkaline soils. The pH change induced by biochar controlled Cd activity more effectively in an acidic environment, while the adsorption properties of biochar dominated in an alkaline environment [13]. In these cases, the remediation effect of the passivating material is dependent on the soil properties. In other cases, however, passivation remediation is dependent on the type and extent of the contamination. For example, a meta-analysis of 489 independent observations on heavy metal immobilization (biochar, phosphate, lime, metal oxides, and clay minerals) showed that phosphate was the most efficient material for Pb immobilization, while lime was the most effective material for fixing Cd. Not only that, but the effect of lime also depended on the activity of Cd in the soil. Lime was most effective when the initial bioavailable Cd concentration was 0.1–1 mg kg^−1^, while the effects of lime and phosphate were similar when the initial bioavailable Cd content was 10 mg kg^−1^ [14]. In addition, metal concentrations can also influence the ability of plants to transport metals. For example, high concentrations of Mn stress may lead to the cell deformation and distortion of plants and reduce the ability of plants to transport Mn [15].

Guizhou Province is located in the karst center of southern China. Karst landforms are distributed throughout Guizhou province except for in the southeast corner, and a large amount of arable land is distributed in the karst mountainous areas. Due to the special geological background, the soil in Guizhou province has a high natural background value of Cd, and the source and control factors are very complicated [16]. In the northwestern part of Guizhou province, the soil not only has a high natural background value of Cd, but is also affected by the history of local zinc smelting [17,18]. The random disposal of lead–zinc smelting residues leads to the transport of Cd in the residues to the soil through hydrological processes and atmospheric deposition [19]. Although the local government has banned zinc smelting, its impact on the environment continues to this day [20]. Soil acidification and nutrient loss may be another problem facing the region. The dissolution of soluble rocks (limestone or dolomite) may form downward fractures, which may lead to downward migration and accumulation of nutrients in the topsoil, resulting in greater nutrient loads (e.g., C, N, P) [21,22]. Nutrient loss means that long-term fertility input is required to maintain crop growth. However, long-term fertilization may lead to massive N deposition, which in turn affects soil exchangeable cation concentrations and exacerbates soil acidification [23]. The combination of exogenous Cd input and soil acidification may lead to the activation of Cd in soil. Under the influence of these factors, the soils in northwestern Guizhou province generally have the characteristics of high Cd concentration, strong activity, and serious acidification. As mentioned above, soil acidification, the initial concentration of bioavailable Cd, and crop type factors may all contribute to the effects of passivation remediation. Hence, applying different passivation remediation materials to Cd-contaminated soils in northwestern Guizhou may need to consider the effects of soil and agricultural characteristics on the remediation effect. However, the current research on this area mainly focuses on the risk assessment of soil Cd contamination, and few studies have compared the effects of different passivation materials.

The northwest of Guizhou province is an important potato-planting base in Guizhou Province and even in China [24]. We selected two potato varieties (Q9 and W5) that are mainly cultivated locally, and we collected local Cd-contaminated soil in situ to carry out pot experiments and compared the application effects of six common materials (lime, apatite, calcite, sepiolite, bentonite, and biochar). We compared the differences in soil Cd activity and the Cd accumulation in the potato plants after application of different passivation materials and discussed the possible mechanism for this in combination with the soil properties and texture. We expect that these experimental results can provide data support and a theoretical basis for the further application of these materials in the study area.

## 2. Materials and Methods

### 2.1. Overview of the Study Area

The study area was Weiningnian County, in the north-western region of Guizhou province, China. The region has a subtropical monsoon climate, with an average altitude of 2200 meters, an average annual temperature of 10.2 °C, and average annual precipitation of 926 mm. Due to the characteristics of small annual temperature difference, short frost-free periods, and long sunshine time, the local climate and geographical conditions are suitable for the development of agriculture, especially crops such as potato and corn. However, under the influence of the high Cd background values and the history of lead and zinc smelting, Cd pollution has become a severe environmental problem for this region [20]. We randomly collected 144 farmland soil samples in Xiaohai Town, located in the center of Weining county, and assessed the pollution risk of six heavy metal elements (Cr, Ni, Cu, Zn, Cd, and Pb) in the farmland soil. The results showed that the concentrations of Cu, Zn, and Cd all exceeded the natural background values, but only the concentration of Cd exceeded the risk-screening value, and the excess multiples ranged from 3 to 10 times (Appendix A). The pollution index (PI) is a method to determine the main pollutants and the risk levels, generally expressed as the ratio of the element content measured in the soil to the standard value of the element. Soils were assessed as medium- or high-risk when the PI value was 3–5 or above 5, respectively [24]. Hence, if we refer to China’s risk assessment standards, the farmland soil in the study area is a typical Cd-contaminated soil, and it is at a medium–high pollution risk level.

### 2.2. Pot Experiment

We evaluated the texture and soil properties of the tested soils before conducting pot experiments. Referring to the “international” size scale, the proportions of sand (2~0.02 mm), silt (0.02~0.002 mm), and clay (0.002 mm) in the soil were 29.6%, 26.5%, and 44%, respectively [25]. The potting soil can be classified as loamy clay. In addition, we determined the physicochemical properties of the potting soil (pH 5.87; organic matter 28.03 g/kg; total nitrogen 1.82 g/kg; available phosphorus 33.6 mg/kg; available potassium 245 mg/kg; alkaline hydrolysis N 111 mg/g; Cd total 1.499 mg/kg).

The pot experiments were conducted in a greenhouse at the Guizhou Academy of Agricultural Sciences from March to August 2019. Each plastic pot (diameter 30 cm; height 40 cm) was filled with 10 kg of air-dried soil, 40 g of organic fertilizer, and 2 g of compound fertilizer. Six passivation materials (lime, calcite, apatite, seafoam, bentonite, and maize biochar) were selected to treat the potting soil, and each treatment was replicated three times. Details of the pot experimental treatments are shown in Table 1. The properties of different passivation materials are shown in Table 2. Two germinated potato blocks were subsequently sown with soil in each pot. An additional 2 g of compound fertilizer was applied during the seedling stage of potato growth (May 2019). Water was added regularly during potato growth to maintain soil moisture at 60% of the field water holding capacity. Destructive sampling of plants and soil was carried out at the time of potato maturity. Soil samples were mixed well and divided into four parts by dividing diagonally, keeping the two diagonal parts, and discarding the remaining two parts until about 1 kg of soil was collected. The collected soil samples were air-dried and passed through 10 and 100 mesh nylon sieves for subsequent analysis. Potato samples were divided into roots, stems, leaves, and tubers. They were dried at 65 °C, weighed, and then ground for analysis.

### 2.3. Determination of Physical and Chemical Properties of Soil

The total concentration of Cd, Fe, and Mn in the soils was digested with HF (1 mL)-HNO_3_ (3 mL)-HClO_4_ (1 mL) by the autoclave in a constant temperature drying oven; the Cd content in potato plants was digested with HNO_3_ (5 mL) by the autoclave. After the soil sample was extracted with water, the pH value of the soil was measured with a pH meter (Sartorius; Germany; PB-10 type); soil organic matter was oxidized with potassium dichromate and titrated with ferrous sulfate; total N was determined by the Kelvin method; total P was determined by H_2_SO_4_-HClO_4_ digestion and molybdate-ascorbic acid method; total K was digested with NaOH and then determined by atomic absorption spectrometer; available P was determined by NaHCO_3_ method; available K was extracted with ammonium acetate and determined by flame spectrophotometry; exchangeable Ca and Mg were determined by atomic absorption spectrophotometry; available Si was extracted with citric acid and determined by molybdenum blue colorimetry; with reference to the ‘international’ size scale, the soil texture category was delineated according to the proportion of the three-grain content of sand (2~0.02 mm), silt (0.02~0.002 mm), and clay (0.002 mm) [25,26]; and the fractionation of Cd was extracted and analyzed by the BCR stepwise extraction method. In step 1, exchangeable Cd (EX-Cd) was extracted from 0.11 mol/L acetic acid (pH = 2.8), in step 2, reducible Cd (RE-Cd) was extracted by 0.1 mol/L hydroxylamine hydrochloride (pH = 2), in step 3, Cd was oxidized by acid-stabilized 30% hydrogen peroxide (OX-Cd) followed by extraction by 1 mol/L ammonium acetate (pH = 2), in step 4, the residue (RS-Cd) from step 3 was digested in HF (1 mL)-HNO_3_ (3 mL)-HClO_4_ (1 mL) [27]; the bioavailability of Cd (DTPA-Cd), Fe (EX-Fe), and Mn (EX-Mn) was extracted by DTPA test (extraction in 5 mM diethylenetriaminepentaacetic acid and 10 mmol/L CaCl_2_) [28]; the concentration of Cd in various fractionations, the EX-Fe, and the EX-Mn were determined by ICP-MS (Inductively Coupled Plasma Mass Spectrometry; Thermo Scientific; iCAP RQ); the data quality was controlled by using blank parallel samples and national standard materials (China). The parallel error of all samples was within 5%, soil standard materials used GSS-5 (Cd recovery rate: 96.9~118.9%), and plant sample standard materials used GSB-1 (Cd recovery rate: 93.1~96.6%).

### 2.4. Data Processing and Analysis

Analysis of variance (ANOVA) and correlation analysis were performed with R software (version 3.6.2, Auckland, New Zealand); for data following normal distribution and with a homogeneous variance, group comparisons were performed by one-way ANOVA followed by Tukey’s HSD test. For skewed data distribution, the Kruskal–Wallisy test was used for ANOVA, and the Wilcoxon test was used for group comparisons; the correlation analysis was conducted in R using the Hmisc package (https://CRAN.R-project.org/package=Hmisc (accessed on 15 August 2022)) and a heatmap was plotted according to the spearman correlation using the package corrplot ( https://cran.r-project.org/package=corrplot (accessed on 18 November 2021)); R package ggplot2 (https://cran.r-project.org/package=ggplot2 (accessed on 3 May 2022)) was used for plotting other figures [29].

## 3. Results and Discussion

### 3.1. Effect of Different Treatments on the Morphology of Cd

Studies have shown that the application of passivation materials can promote the stabilization of Cd in soil, thereby reducing the activity of Cd in soil. Therefore, we compared the effects of different passivation materials on the morphology of Cd in the soil, and the results are shown in Figure 1. Despite morphological variation between the treatments, the overall trend was EX-Cd > RE-Cd > RS-Cd > OX-Cd. Metal cations in the soil may be present in several chemical forms, i.e., as easily exchangeable ions, as ions combined organic, as occluded by or co-precipitated with metal oxides or carbonates or phosphates and secondary minerals, and as ions in crystal lattices of primary minerals [30]. These forms correspond to exchangeable-carbonates (EX-Cd), reducible-iron/manganese oxides (RE-Cd), oxidizable-organic matter (OX-Cd), and residual-crystal lattices (RS-Cd), respectively [27]. Hence, the high percentage of EX-Cd indicates that the Cd in soil was easily migrated and transformed, and then enriched in crops. However, the result of the ANOVA showed the application of LM, CA, and AP significantly reduced the soil EX-Cd (Tukey’s HSD; *p* < 0.05; Figure 1). The EX-Cd reduction was accompanied by an elevated level of RE-Cd, OX-Cd, RS-Cd. A strong relationship between the mobility of metals and the chemical forms has been reported in the literature. The observed shift in the morphology indicates that the LM, AP, and CA induced a transformation from the exchangeable form of Cd to a stable one. The results were in line with the former studies, in that the application of the LM, AP, and CA enhanced the soil pH and stabilized the level of Cd in soil [8,31,32,33]. One possible mechanism is that the ion exchange (Metal/Ca) and co-precipitation (≡PO_4_O-Metal) prompts Cd^2+^ to absorb on the surface of the materials or to form stable compounds [7,31,34].

In addition, the appreciable changes in the morphology of the Cd were not observed after the application of BN and SP. This trend is not consistent with previous studies that showed that the application of BN and SP decreased Cd activity in soil [6,31]. This could be because the adsorption of Cd^2+^ by clay minerals is strongly dependent on pH and ionic strength. However, at low pH, the absorption of Cd^2+^ is dominated by outer-sphere surface complexation and ion exchange, which is inhibited under acidic conditions [35,36,37].

### 3.2. Effect of Different Treatments on Cd Concentrations in Potato Tubers

To evaluate the effect of different materials on the Cd content of the potato tubers, the potatoes were co-cultivated with different passivation materials (Figure 2). The results showed that LM, AP, and CA significantly decreased the Cd enrichment of the potato tubers (Tukey’s HSD; *p* < 0.05). The results from the DTPA method were consistent with the BCR method, in that the application of LM, AP, and CA significantly decreased DTPA-Cd (Tukey’s HSD; *p* < 0.05). Following LM, AP, and CA neutralization, the pH of the soils was neutral or close to neutral (6.3–7.4). Soil pH is closely related to the surface chemistry of soil colloids and is an important factor affecting variable charge. The amount of soil variable negative charge and cation exchange increases with the enhancement of soil pH, which in turn promotes the formation of stable complexes of metal ions [38]. The increase in soil pH also facilitates the release of OH^-^ from the soil, which in turn promotes the formation of carbonate or hydroxide precipitates from heavy metal ions [39]. On the one hand, the increase in soil pH is conducive to the formation of hydroxide, phosphate, and carbonate precipitation of Cd in soil. On the other hand, this enhances the ion exchange between Cd^2+^ in soil and surface Ca^2+^ of calcite, apatite, and other mineral materials, which in turn promotes the adsorption and precipitation of Cd^2+^ by calcite and apatite.

Notably, BC incited a decrease in the EX-Cd concentration but had no effect on DTPA-Cd (Figure 1 and Figure 2). One probable reason is that the extraction agent with a pH of 7.3 prevents calcium carbonate from dissolving and releasing closed Cd, and thus the DTPA extraction method was more suitable for calcareous and alkaline soils [40]. As prior studies have shown, the three extractants DTPA-TEA, TCLP, and MgCl_2_ extracted different concentrations of Cd from the soil [40]. Therefore, the remediation potential of biochar cannot be denied. In fact, biochar is an excellent sorbent for Cd^2+^ and a solid residue of biomass pyrolysis under low oxygen conditions, which has a poly-aromatic and microporous structure, diverse surface functional groups, and a strong cation exchange capacity [41,42]. Although the DTPA-Cd had no change under the BC treatment for methodological reasons, BC remains a potential material. The decrease in EX-Cd and Tuber-Cd under BC treatment could support this suggestion (Figure 1).

Additionally, we did not observe significant differences in the morphology of the Cd and Tuber-Cd after applying BN and SP (Figure 2). A previous study showed that BN and SP may affect the activities of peroxidase, superoxide dismutase of crop leaves, and the activity of soil bacteria and fungi, which influences the absorption of Cd by plants [6]. Therefore, we further analyzed the difference in Cd concentration in the potato plants after application of passivation materials.

### 3.3. Effects of Different Treatments on Cd Concentration in Potato Plants

Another factor affecting the Cd concentration in potato tubers is the distribution of Cd among potato plants. Previous studies have provided convincing evidence that Cd is not transferred directly from the basal root to the xylem of the tuber, but takes a pathway involving the phloem [43,44]. Hence, Cd in potato tubers is more likely attributable to the partitioning of Cd within the plant than to the direct uptake of Cd from the soil by the potato pericarp [44]. We compared the differences in Cd concentrations in different tissues of potato after the application of passivated materials to assess the effect of passivated materials on Cd transport in potatoes (Figure 3). The results showed that the concentration of Cd in different potato tissues showed the trend of stem > leaf > tuber. However, Cd in the Q9 roots was more easily transported to the stem than in W5. Previous studies have shown that Cd in soil can be enriched to the aerial part through the potato root system and migrate in the xylem and phloem of the plant. In this process, potato stems may act as important ‘reservoirs’ and ‘hubs’ and may regulate the migration of Cd in various tissues of the potato. Subsequently, Cd in potato stems may be distributed to the leaves and migrate through the phloem to the potato tuber [43].

In contrast, the application of passivated materials caused varying degrees of Cd concentration in various tissues of the potato (Figure 3). The effects of LM, CA, and AP on Cd concentration in all parts of Q9 and W5 showed a similar pattern, and that their application resulted in a significant decrease in Cd concentration in all tissues of the potato compared to CK. This corresponds to our previous findings that LM, CA, and AP could effectively reduce the availability of Cd in soil, which in turn inhibits the accumulation of Cd in different potato tissues. By reducing the total ‘flux’ of Cd into the potato plants, LM, CA, and AP effectively controlled the accumulation of Cd in the potato tubers. However, the variation in Cd from root to stem in both Q9 and W5 showed an increasing trend after BN and SP application, suggesting that the application of these two materials may have enhanced the translocation of Cd from root to above ground. Studies have shown that sepiolite can retain water and can be used as a water storage media to offset soil moisture deficiencies, or to improve the soil microbial habitat and soil structure by affecting soil enzyme activity [45]. Similarly, bentonite is considered ideal for improving soil texture and is good for improving the soil moisture retention and fertility [46]. Hence, we assumed that the application of bentonite and sepiolite may have improved the soil environment and thus improved the transport of mineral elements between soil–plant systems. As mentioned above, the accumulation of Cd in potato tubers is related to the distribution of Cd among the potato parts. Therefore, this result may partly explain why the Cd concentration in potato tubers treated with BN and SP was relatively higher than in the other treatments. It is noteworthy that the BC treatment enhanced the translocation of Cd from Q9 roots to stems, while W5 showed the opposite pattern, which is consistent with the translocation characteristics of the two species themselves. The accumulation of Cd in all tissues of W5 was higher than that of Q9 when no passivation material was applied, and Cd from the roots of W5 was not easily transported to the shoots of potato. This result suggests that the effect of BC application was closely related to the genotype of potatoes. An earlier study showed that the genotype explained most of the variation (40–70%) in rice biomass and nutrient uptake, while biogenic carbon effects explained only 7% of this variation [47]. Another study indicated that biochar treatment reduced Cd accumulation in various parts of most wheat varieties, but caused Cd enrichment in different tissues of individual varieties [48]. On the one hand, plants with different genotypes can have different root characteristics. Differences in root environment (e.g., root-secreted organic acids) may affect Cd activity in the soil and hence the accumulation of Cd by plants [49]. On the other hand, differences in plant genotypes can also influence the distribution of heavy metals to plant organs and stressful reactions to environmental factors [48]. In summary, the application of LM, CA, and AP effectively reduced the Cd concentration in the potato tubers through the inhibition of Cd enrichment in all tissues of the potato. However, the practical application of BN, SP, and BC requires more consideration of the distribution of Cd among different potato tissues and the effect of the potato genotype.

### 3.4. Correlations between Soil Cations and Availability of Cd

To explain the passivation mechanism of different materials, we contrasted the differences in fertility, texture, and cationic exchange capacity of the soils after the application of different materials. The influence of fertility and mechanical composition was excluded first because we found no significant difference between CK and treatment (Coefficient of variation from 2% to 15%; Appendix A). Apart from Ca, we also found no differences in the total amount of cations in the soils (Coefficient of variation from 3% to 9%; Appendix A). However, significant differences were observed between LM, AP, CA, and CK for EX-Ca, EX-Mg, EX-Si, EX-Fe, EX-Mn, and the total concentration of Ca (Kruskal–Wallis and pairwise-Wilcox; *p* < 0.05; Figure 4). The application of LM, AP, and CA significantly increased the Total-Cd, EX-Si, and EX-Ca, as well as decreased the EX-Fe and EX-Mn. The Total-Ca change patterns corresponded to the soil pH (Figure 2). It can therefore be assumed that the Ca component may be the main factor controlling the soil acidity and alkalinity. This result is consistent with earlier findings that showed that alkalinity associated with calcium deposition neutralizes soil acidity [50]. The main reason is that Ca compounds not only produce OH^-^ and HCO^3-^ by hydrolysis to neutralize H^+^ in the soil, but also replace the H^+^ and Al^3+^ adsorbed onto the soil colloid to increase the soil alkaline saturation [38]. In addition, the ratio of pH to cations in soils might explain why there was no change in nutrient activity in the soils. Previous studies have shown that increasing the concentration of Ca inhibited the accumulation of DOC in soil filtrate at neutral pH, but had no effect on DOC or DON in soil filtrate at acidic pH [51]. Another study suggested that P solubility in soil is related to soil pH, iron, or aluminum oxides, while the Ca contained in biochar may coprecipitate with P [52].

The results from the correlation analysis showed a significant negative correlation for Tuber-Cd and EX-Mg (R2 = 0.21; *p* < 0.05; Figure 5b), whereas there was a significant positive correlation for Tuber-Cd and EX-Cd (R2 = 0.36; *p* < 0.01; Figure 5c). In contrast to other cationic elements, there was no significant correlation between EX-Mg and EX-Cd. Hence, the effect of Mg on the Cd content of the potato tubers may not result from changes in the availability of Cd in the soil (Figure 5a). We therefore assumed that EX-Mg would have a direct effect on Tuber-Cd. As indicated in previous studies, magnesium application is beneficial for inhibiting Cd uptake by plants [53,54]. Previous research also suggests that Mg inhibits the accumulation of Cd in plant roots or the translocation of Cd to stems through competitive adsorption with Cd [55]. This may be the result of the competitive adsorption of Mg and Cd by plants, but additional monitoring experiments are needed to verify this [56,57]. Furthermore, a significant positive correlation was found among EX-Si, EX-Fe, EX-Mn, EX-Al, and Cd (Figure 5a). This may therefore suggest that these parameters indirectly reduced Tuber-Cd by affecting EX-Cd. This is consistent with a previous study that showed that an increase the soil Si and Ca composition may increase soil pH and inhibit soil Cd migration by co-precipitation or surface complexation [58]. In addition, the Fe-Mn (oxyhydro) oxide fraction was found to be an important factor in Cd solubility [59].

## 4. Conclusions

The application of LM, CA, and AP had positive effects on the soil properties such as soil pH and cations and effectively reduced the activity of Cd in the soil and the accumulation of Cd in the potato plants. In contrast to previous studies, the application of SP, BN, and BC had no significant impact on the soil Cd availability. The application of SP and BN enhanced the accumulation of Cd from potato roots to shoots, while the effect of BC on the Cd content of the potato plants varied by different potato varieties. In contrast, calcareous materials (LM, CA, and AP) are more suitable for soils in areas affected by lead and zinc mines with severe acidification and high Cd activity. Although the application of calcareous materials effectively reduced the Cd concentration in the potato tubers, the value was still higher than the Chinese food safety limit (0.1 mg/kg). Further research can combine foliar fertilizers and various passivation materials to further improve the repair efficiency. However, the effect of potato genotype should be considered when applying organic materials such as biochar.

## Figures and Tables

**Figure 1 ijerph-19-11736-f001:**
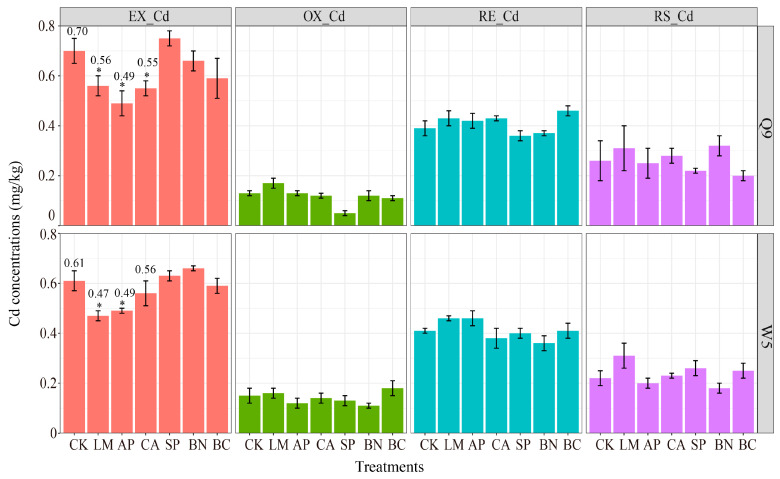
Effect of the morphology of Cd in Cd-contaminated soil after the application of different materials; Q9 and W5 represent the potato cultivar; EX-Cd, RE-Cd, OX-Cd, and RS-Cd represent exchangeable, reducible, oxidizable, and residual Cd, respectively; asterisks represent a significant difference from control (CK) (*p* < 0.05, ANOVA; pairwise-Tukey’s HSD). CK, Control; LM, Lime; CA, Calcite; AP, Apatite; SP, Sepiolite; BN, Bentonite; BC, Biochar.

**Figure 2 ijerph-19-11736-f002:**
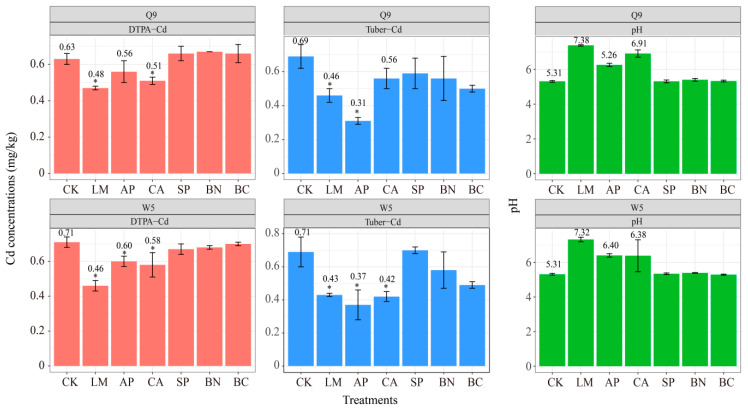
Effect of different treatments on pH and bio-available Cd (DTPA-Cd) of soils, and Cd concentrations in potato tubers (Tuber-Cd); asterisks represent a significant difference from control (CK) (*p* < 0.05, ANOVA; pairwise-Tukey’s HSD). CK, Control; LM, Lime; CA, Calcite; AP, Apatite; SP, Sepiolite; BN, Bentonite; BC, Biochar.

**Figure 3 ijerph-19-11736-f003:**
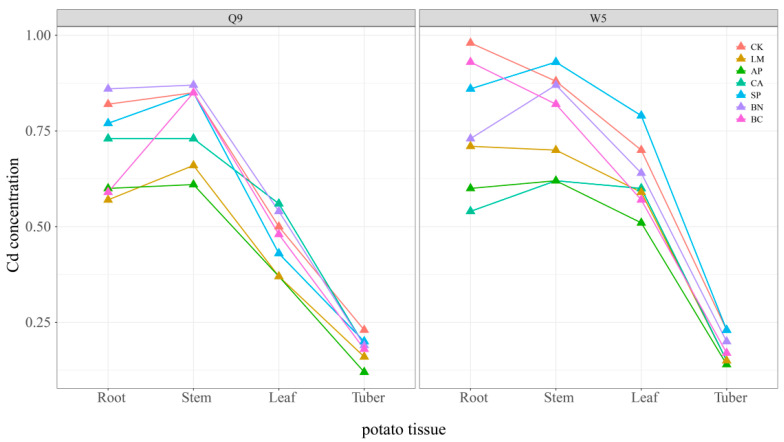
Effects of different passivation materials on Cd content in potato roots, stems, leaves, and tubers. For a clearer expression, we pre-logarithmically processed the data. CK, Control; LM, Lime; CA, Calcite; AP, Apatite; SP, Sepiolite; BN, Bentonite; BC, Biochar.

**Figure 4 ijerph-19-11736-f004:**
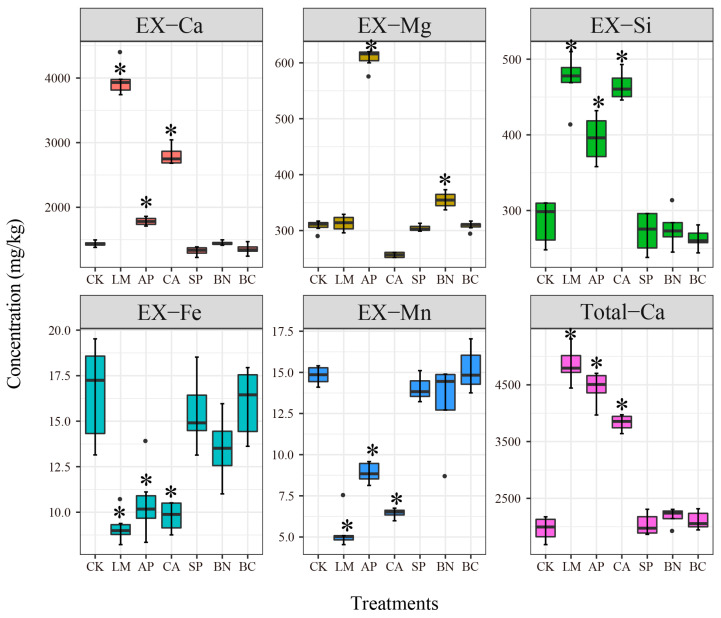
Effect of different treatments on the soil cations; Total-Ca, EX-Ca, EX-Mg, EX-Si, EX-Fe, and EX-Mn represent total concentrations of Ca and exchangeable Ca, Mg, Si, Fe, and Mn, respectively; asterisks represent a significant difference from control (CK) (*p* < 0.05; Kruskal–Wallisy; pairwise-wilocx).

**Figure 5 ijerph-19-11736-f005:**
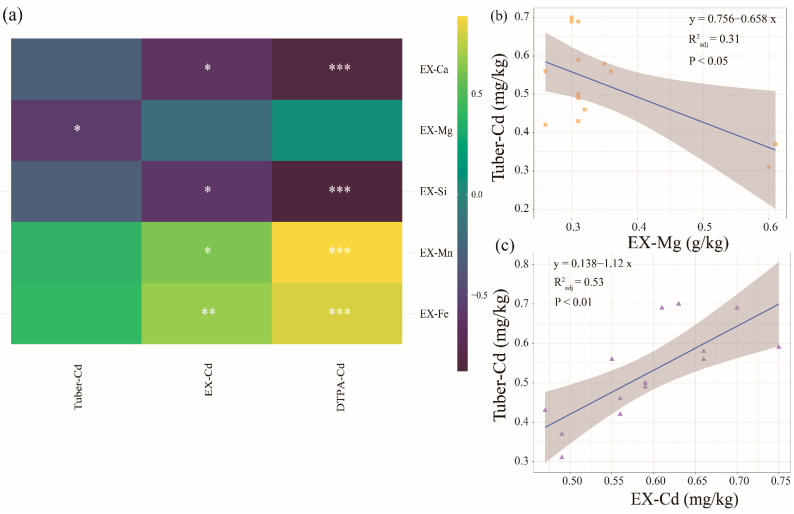
The correlation between the passivation effects and the soil cations. (**a**) The heatmap represents the correlation of parameters (Spearman correlation coefficient). The color difference represents the correlation coefficient and the asterisk indicates that the correlation is significant, “*” (*p* < 0.05), “**” (*p* < 0.01), “***” (*p* < 0.001); (**b**) The linear regression of EX-Mg versus Tuber-Cd. (**c**) The linear regression of EX-Cd versus Tuber-Cd.

**Table 1 ijerph-19-11736-t001:** Pot experiment treatment.

Material	Treatment	Potato Cultivar	Dosage (Material Weight/Soil Weight)
Control	CK	Q9	0%
Control	CK	W5	0%
Lime	LM	Q9	0.4%
Lime	LM	W5	0.4%
Calcite	CA	Q9	0.4%
Calcite	CA	W5	0.4%
Apatite	AP	Q9	1.4%
Apatite	AP	W5	1.4%
Sepiolite	SP	Q9	0.35%
Sepiolite	SP	W5	0.35%
Bentonite	BN	Q9	1.4%
Bentonite	BN	W5	1.4%
Corn biochar	BC	Q9	0.4%
Corn biochar	BC	W5	0.4%

**Table 2 ijerph-19-11736-t002:** Chemical characterization of materials.

Materials	Main Ingredient	pH	Cd/(mg/kg)	Abbr.
Lime	CaO	12.38	1.176	LM
Apatite	CaO (55.38%); P_2_O_3_ (42.06%)	7.62	0.232	AP
Calcite	CaO (56.03%); CO_2_ (43.97%)	8.9	0.730	CA
Sepiolite	SiO_2_ (55.65%); MgO (24.89%)	7.69	0.031	SP
Bentonitic	SiO_2_ (66.7%); Al_2_O_3_ (28.3%)	8.83	0.291	BN
Corn biochar	/	7.77	0.214	BC

## Data Availability

The data presented in this study are available in article and Appendix A.

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
