# Peer review of "Calcareous Materials Effectively Reduce the Accumulation of Cd in Potatoes in Acidic Cadmium-Contaminated Farmland Soils in Mining Areas"

_ijerph, 2022, doi:10.3390/ijerph191811736_

Round 1
Reviewer 1 Report
ijerph-1901020
Title: Calcareous materials effectively reduce the accumulation of Cd in potatoes in acidic cadmium-contaminated farmland soils in mining areas
I have reviewed your MS thoroughly with great interest. The hypothesis is fine but the planning of your experiment looks very outdated and I could not find anything new compared to that with the existing literature. The abstract and introduction need to be greatly revised. And the current data is not sufficient to reach a logical conclusion. So, I will advise to add some more data in your MS. Besides, in its current state, the English should be modified to meet the journal’s required. You may check your MS for grammar, style and syntax carefully. Detailed comments are as follows.
The abstract and introduction need to be greatly revised. Line11-16 should be modified.Why cations and pH positive effects of availability Cd in soil and uptake of Cd in potatoes in this study? LM, AP and CA has also been reported in previous studies, where is the innovation of this paper? The introduction does not provide a good review of the previous literature and does not distill key scientific questions. The aim of this study should be rewritten.
The risk index of Cd contamination in the study area should be assessed.
Why choose Q9 and W5 two varieties? Haven't seen the comparative analysis of Q9 and W5 in the results and discussion?
The specific experimental conditions for each treatment group in the study is unclear, such as materials proportion? Does proportion of different materials affect the experimental results?
Detailed soil physicochemical properties after adding different materials should be compared in detail.
The effects of passivating materials on soil microbes and rhizosphere effects cannot be ignored.
The effects of passivation materials on the physiological and biochemical properties of potatoe is not discussed, and the Cd distribution in potato various and the ability of potatoes to accumulate and transport Cd should also be discussed.
Charts need to be checked carefully. For example, Line 193 Figure 2 DTPA-Cd, Tuber-Cd, and pH should not share a Y-axis description.
There are many language errors in the manuscript, Such as Line 166 “To assess the application of these materials to agricultural lands.”.
A large proportion of obsolete reference materials need to be rechecked. And the format of reference should be carefully checked. For example, Line 287, Line 304, Line 308, Line 317, Line 345, Line 378, Line 394.
Reviewer 2 Report
In this study, a pot experiment was conducted to select the best passivation material for application and actual production by comparing the effects of six materials (Lime; Apatite; Calcite; Sepiolite; Bentonitic; and Biochar) for two potato varieties. The authors are to be commended for undertaking such work. However, there are concerns with the current version of the manuscript.
(1) Similar pot experiments have been reported. The author should put forward the innovation of this paper on the basis of summarizing previous studies. What are the key problems of poor field application? How the author cut in was not explained. It is better to add related references and contents in the Introduction section.
(2) The “design of the pot experiments” part should be revised thoroughly (lines 90-96). The test design shall introduce the test factors, the level of factors and the content of test treatment in detail. But these details are missing.
(3) The discussion part needs a substantial revision for a better logical and clear description of the main points that the author want to address. The results cannot be summarized in a descriptive way in this section and the mechanism of the difference should be analyzed and reasoning. It is necessary to analyze the relationship between pH and cations.
Reviewer 3 Report
The analyzed document “Calcareous materials effectively reduce the accumulation of Cd in potatoes in acidic cadmium-contaminated farmland soils in mining areas “presents a survey ,at small pot scale, about the possibility of recovering a soil contaminated with cadmium through the use of different regenerative materials. The study presents an evident agronomic interest, perhaps focused at regional scale, since the soils are local and not well characterized, the potato varieties used are also local, and the conditions are difficult to reproduce in other experiments.
The data presented are interesting from the agronomic and scientific point of view, although the study in general lacks originality since the recovery of soils contaminated with heavy metals using calcareous materials is a practice that has been used successfully in many areas of the world.
I believe that the study as a whole, match with the topic of the journal and can be published if some aspects listed below are improved:
line 18: “biochar” is missing an r.
line 20: the sentence it is not well understood because the figures provided 0.42% W/W , are refered to the pH, to soil cations or to calcium or extractable cadmium??.
Line 30: “control” is misspelled with a space.
Line 44: there are some references missing, from my point of view (i.e. browse sugar foam).
Line 75 and following: One of the main points of improvement that should be included in this manuscript is the description of the soil. Data as important as granulometry are missing. It would be necessary to use USDA or FAO international description criteria that will lead us to know the soil in its different horizons and its classification. A deep knowledge of the soil will make it easier for us to propose new remediation systems, such as mixing deeper horizons or trying to bury the contaminated surface horizons.
Line 84: the hydrolyzed or hydrolyzed nitrogen parameter is not commonly used.
Line 92: Soil moisture maintained by irrigation is not well defined. It refers to 60% of the useful water??? field capacity minus the wilting point???.
Line 93 and following: it is not clear what amount of sample or how that soil sample was taken to carry out the tests.
Line 105: the reference for analytical methods is local. It should be clarified if this reference is based on common methods used internationally.
Line 137: the order suggested in the text does not coincide with the order that appears in the figure, particularly between the last two quantities.
Line 151: what does “Me” mean?
Line 161: figure 1 is not completely self-explanatory since it would be convenient to clarify what the initials of each of the treatments mean.
Line 194: figure 2; The first two figures on the left are very similar to those in Figure 1.
Line 210 and following: as can be seen throughout the discussion, almost all the data provided are consistent with most of the studies previously carried out, which gives an idea of ​​its weak novelty.
line 232: metal transport proteins in plant roots are used, in most cases, to transport all metals that have a similar charge and ionic radius; not only magnesium and cadmium, but also copper, iron, zin,c etc reference: Marschner, P. (2012). Nutrition of higher plants. 3th edición. Ed. Elservier.
Line 243: Figure 4b; Taking into account the low value R2 = 0.31, it is very risky to draw the consequence that the increase in magnesium decreases the cadmium content of the potato. In any case, it would not be a strange consequence since in many cases calcium and magnesium are provided together in remediation materials such as calcium and magnesium carbonate (dolomite).
Line 249: from my point of view, the conclusions could be reformulated and made clearer.
Round 2
Reviewer 1 Report
Manuscript: Calcareous materials effectively reduce the accumulation of Cd in potatoes in acidic cadmium-contaminated farmland soils in mining areas (IJERPH -1901020)
The manuscript submitted by Gong is suitable for IJERPH, and some interesting results were showed. However, still some minor problems were existed. Firstly, the English needs to be carefully checked. Furthermore, more details were given as bellows.
The lack of line numbers in the full text makes it difficult to review.
Page 1 The in-situ chemical fixation method reduces the activity of heavy metals in soil by adding chemical modifiers to the soil. “to the soil” should be deleted.
Page 2 some sentences should be quoted, and some latest literature reports should be considered in the introduction, such as International Journal of Environmental Research and Public Health 2022, 19(16), 10353; Environmental Science and Pollution Research 2022, 29, 39017–39026.
References need to be double-checked. For example,
“46. Zhang, G.; Fukami, M.; Sekimoto, H. Genotypic Differences in Effects of Cadmium on Growth and Nutrient Compositions in Wheat. null 2000, 23, 1337–1350, doi:10.1080/01904160009382104. ” should be
“46. Zhang, G.; Fukami, M.; Sekimoto, H. Genotypic Differences in Effects of Cadmium on Growth and Nutrient Compositions in Wheat. J Plant Nutr 2000, 23(9), 1337–1350, doi:10.1080/01904160009382104.”.
And some references are too old, some recent literature reports should be considered.
Reviewer 2 Report
The revised manuscript is quite interesting and well organized, I think this manuscript is suitable for publication.
Author Response
Once again, we thank the reviewers and editors for the valuable comments on the manuscript, which were crucial to improving the manuscript.
Reviewer 3 Report
From my point of view, the manuscript has been improved by the authors and most of the issues raised in the review have been satisfied. I only miss an improvement proposed in the revision on the analytical methods used internationally in soil. I trust the editor's judgment with the explanations provided by the authors since I am not a specialist in analytical chemistry.
The "L" in the formula H2SO4-HCLO4, does not correspond to any chemical element (determination of P).
